# Identification of Immune Cell Components in Breast Tissues by a Multiparametric Flow Cytometry Approach

**DOI:** 10.3390/cancers14163869

**Published:** 2022-08-10

**Authors:** Luigi Coppola, Giovanni Smaldone, Massimiliano D’aiuto, Giuseppe D’aiuto, Gennaro Mossetti, Massimo Rinaldo, Simona Verticilo, Emanuele Nicolai, Marco Salvatore, Peppino Mirabelli

**Affiliations:** 1IRCCS SYNLAB SDN, Via E. Gianturco, 80143 Naples, Italy; 2Clinica Villa Fiorita, Via Filippo Saporito, 24, 81031 Aversa, Italy; 3Pathological Anatomy Service, Maria Rosaria Clinic, Pompei, 80045 Naples, Italy

**Keywords:** tumor microenvironment, TILs, breast cancer, deep flow cytometry

## Abstract

**Simple Summary:**

The tumor microenvironment in breast cancer plays important roles in tumor development and treatment response, giving important information critical for disease management. Today, an analysis of the tumor microenvironment is included in routine histopathologic reporting for practical clinical application. This manuscript aimed to deepen the study of the tumor microenvironment, analyzing the immune cells in breast tumoral and benign pathologies. Indeed, using a deep immunophenotyping approach by flow cytometry, we have studied the immune cells at the level of breast tissues, identifying different immunophenotyping that could be useful in the diagnosis and follow up of breast pathologies. As possible targets are continually being discovered in the tumor microenvironment, a future approach to breast cancer diagnosis and therapy could likely combine cancer cell elimination and tumor microenvironment modulation.

**Abstract:**

Immune cell components are able to infiltrate tumor tissues, and different reports described the presence of infiltrating immune cells (TILs) in several types of solid tumors, including breast cancer. The primary immune cell component cells are reported as a lymphocyte population mainly comprising the cytotoxic (CD8+) T cells, with varying proportions of helper (CD4+) T cells and CD19+ B cells, and rarely NK cells. In clinical practice, an expert pathologist commonly detects TILs areas in hematoxylin and eosin (H&E)-stained histological slides via light microscopy. Moreover, other more in-depth approaches could be used to better define the immunological component associated with tumor tissues. Using a multiparametric flow cytometry approach, we have studied the immune cells obtained from breast tumor tissues compared to benign breast pathologies. A detailed evaluation of immune cell components was performed on 15 and 14 biopsies obtained from breast cancer and fibroadenoma subjects, respectively. The percentage of tumor-infiltrating T lymphocytes was significantly higher in breast cancer patients compared to patients with fibroadenoma. Infiltrating helper T lymphocytes were increased in the case of malignant breast lesions, while cytotoxic T lymphocytes disclosed an opposite trend. In addition, our data suggest that the synergistic effect of the presence/activation of NK cells and NKT cells, in line with the data in the literature, determines the dampening of the immune response. Moreover, the lymphocyte-to-monocyte ratio was calculated and was completely altered in patients with breast cancer. Our approach could be a potent prognostic factor to be used in diagnostic/therapeutic purposes for the improvement of breast cancer patients’ management.

## 1. Introduction

Breast cancer remains the principal cause of death among women, with 2,261,419 newly diagnosed cases in 2020, of which 684,996 were mortality cases [1]. Based on gene expression and molecular features, breast cancer is classified into four subtypes and includes luminal A, luminal B, HER2-enriched, and triple-negative (TNBC). Current treatments are based on the clinico-pathological features, including chemotherapy, hormone therapy or other innovative therapies, radiotherapy, and/or surgical approaches. The tumor microenvironment components, such as fibroblasts, tumor-associated macrophages (TAM), endothelial cells, mesenchymal stem cells (MSCs), and immune cells, play a pivotal role in tumor development, progression, and metastases [2]. Several emerging lines of evidences suggest that tumors are not only composed of malignant cells but have a highly altered surrounding stroma. Indeed, the tumor microenvironment is recognized as a critical factor in tumor development and a considerable parameter for response to treatment.

The presence and infiltration of immune cells at the level of tumoral formation predict an improved prognosis in many different tumor types, such as lung and colon, including breast cancer [3,4]. Regarding breast cancer, several studies showed tumor-infiltrating lymphocytes (TILs) as a favorable prognostic biomarker able to predict response to therapy [5]. In particular, CD4+ T-helper 1 (Th1) cells facilitate antigen presentation through cytokine secretion and activation of antigen-presenting cells. CD8+ cytotoxic T-cells are essential for tumor destruction [6].

However, type 2 CD4+ T-helper cells (Th2), including Forkhead box P3 (FOXP3) CD4+ regulatory T-cells, inhibit CD8+ cytotoxic T-cell function, promoting an anti-inflammatory immune response that could stimulate tumor growth [7]. In breast cancer, FOXP3 expression was associated with worse distant metastases-free survival, and the risk increased with increasing FOXP3 immunostaining intensity [8]. The breast cancer tumor microenvironment promotes the tumor progression by two main molecular mechanisms: (i) releases immune-suppressive factors that make the antigen presentation difficult and that have a negative impact on the immune response; and (ii) blocking endogenous immune checkpoints that determine immune responses after antigen activation [9].

Today, the scientific community is increasingly interested in the breast cancer microenvironment as a prognostic factor to evaluate the possible potential therapeutic targets against stromal components to promote anti-tumor action. To this aim, breast cancer immunotherapy consists of vaccines that target immune responses to tumor-associated antigens and also antibodies that block the checkpoint and inhibit immune suppression by targeting the key pathways mediated by cytotoxic T lymphocyte-associated antigen 4 (CTLA4), programmed death 1 (PD-1), and programmed death ligand 1 (PD-L1). These approaches are currently under investigation as potential strategies for the treatment of breast cancer [10]. Generally, normal breast tissue does not contain infiltrating immune cells [11]; conversely, when neoplastic transformation occurs, immune and stromal cells invade the neoplastic tissue, and their presence can be associated with the different breast cancer subtypes [7,12]. The role of immune cell components in cancer development and the cancer microenvironment represents an object of intense debate in the scientific medical literature. Specifically, in the case of breast cancer, a more favorable clinical outcome was reported when higher numbers of TILs were detected in hematoxylin and eosin (H&E)-stained histological slides via light microscopy [13,14,15,16]. In vitro studies have demonstrated the capacity of T cells isolated from breast tumor tissues to kill tumor cells, underlining they were capable of cytolytic activity [17]. In clinical practice, the presence of TILs is evaluated in hematoxylin and eosin (H&E)-stained histological slides via light microscopy by an experienced pathologist. According to the International TILs Working Group 2014, the BC immune components are divided into stromal compartment TILs and intra-tumoral compartment TILs [18]. The stromal TILs infiltrate the adjacent stromal tissue into the tumor cells, while the intertumoral TILs actively infiltrate tumor cells with direct contact. The distribution of TILs in breast tumors assessed by H&E is challenging due to different levels of tissue sectioning and TILs distribution, such as focal or multifocal. Moreover, the occurrence of a gradient between areas with high and low TILs, and finally, the amplitude of the stromal area, can influence the TILs evaluation by the pathologist. In this scenario, the assessment of the immune cells in the breast cancer microenvironment may be not sufficiently evaluated by classical H&E staining. Therefore, other techniques, such as specific IHC staining, can be more affordable to define the TILs immune components [19,20]. However, an additional way to describe the immune cell components could be a multiparametric flow cytometry (FCM) approach. Indeed, despite FCM having been extensively used for the characterization of normal and pathological lymphocytes in liquid specimens such as peripheral blood and bone marrow samples [21,22,23], fewer reports evaluated its usefulness for defining immune cell subpopulations in solid cancer tissues [24]. These protocols provide for an optimization phase of the breast tissue digestion protocol based on the experimental objective. In general, for solid tumors the experimental protocols foresee a mechanical digestion phase, through the use of scalpels, and an enzymatic digestion phase that favors the obtaining of a sample for single-cell analysis [25]. In the present manuscript, we have analyzed the immune cells components (T-lymphocytes, NK-lymphocytes and monocytes) obtained from breast tumor tissue compared with those obtained from benign breast pathologies using an FCM approach to provide in detail the profile of immune cells residing in malignant and benign breast lesions.

## 2. Materials and Methods

### 2.1. Patient Samples

The present study comprised 29 female patients who underwent breast surgery at the Clinica Villa Fiorita S.p.A, between June 2020 and May 2022 (Aversa, Italy). The study was approved by the Ethics Committee of IRCCS Pascale (Naples, Italy) with reference number 3/19 approved on 29 May 2019. All methods were performed in compliance with standard operating procedures and in accordance with the Declaration of Helsinki and each patient participated in the study by signing written informed consent. Table 1 summarizes the characteristics of the patients included, which was 15 women with malignant breast cancer and 14 women with benign fibroadenoma.

### 2.2. Hematoxylin and Eosin (H&E) Staining

Sections (4 μm) were deparaffinized with 2 changes of xylene for 10 min each. The breast cancer sections were hydrated by passing through a decreasing alcohol series (100, 95, and 70%). Slides were stained in hematoxylin for 10 min at room temperature and then washed under running tap water for 5 min. Acidified alcohol (1%) was used for differentiation (1% HCl in 70% alcohol) for 2 min. Sections were washed under running tap water until the sections were blue again by dipping in an alkaline solution followed by another tap water wash. Then the sections were stained in 1% eosin Y for 5 min at room temperature. Sections were washed in tap water for 3 min and dehydrated in increasing concentration of alcohol and cleared in xylene.

### 2.3. Breast Tissue Digestion and Flow Cytometry Analysis

Flow cytometry was used to evaluate the immune cells at the level of the tumor microenvironment. For the isolation of the immune cell components, a protocol established in our laboratory, based on previously used methods, was applied [26,27]. We used freshly resected tissue that was manually minced with a scalpel blade and then incubated overnight at RT in a shaker table bed (100 rpm/min) with 1.0 mg/mL collagenase A (Roche Diagnostic GmbH, Basel, Switzerland) and 10 µg/mL hyaluronidase from bovine testes, Type I-S (Sigma Aldrich, Saint Louis, MO, USA) diluted in DMEM Medium (Gibco; Thermo Fisher Scientific, Milan, Italy). Subsequently, a single-cell suspension was prepared by filtering through 70-μm nylon strainers (BD Biosciences, Franklin Lakes, NJ, USA). For surface staining, cells were then washed twice with PBS containing 2% FBS (Gibco; Thermo Fisher Scientific, Inc.) and then labeled together with the manufacturer’s suggested dilutions (incubation, 30 min at R.T.) of the following antibodies: CD45 KO, CD8-FITC/CD4-PE/CD3-ECD Antibody cocktail, HLA-DR-PE-Cy5, CD19 PC7, CD 326 (EpCAM) APC, CD56 APC Alexa Fluor 700, CD 14 APC Alexa Fluor 750, and CD16-PB. All the antibodies were obtained from Beckman Coulter, Milan, Italy. For the evaluation of T-reg, we used Duraclone IM T-reg tube (B53346, Beckman Coulter). According to the Beckman–Coulter instruction for the FMO controls, we used VersaComp antibody capture beads in association with the single Compensation Kit provided in the DuraClone IM Treg kit. These beads allow the positive and negative signals associated with the background to be identified separately. The CytoFLEX instrument was used to perform the experiment. CytExpert software 2.4 (Beckman Coulter, Brea, CA, USA) was used to obtain the data. Kaluza Analysis software was used to analyze and characterize the immune cells obtained from the cytometry data.

### 2.4. Statistical Analysis

All statistical analyses were conducted using GraphPad 6 software. A value of *p* < 0.05 was considered to indicate a statistically significant difference. All experiments were repeated three times.

## 3. Results

### 3.1. Breast Tissue Morphological Evaluation

Morphological analysis of H&E-stained breast tissue slides allowed us to define the degree of TILs, as presented in Figure 1 and Appendix A. An experienced pathologist reviewed the slides and evaluated the lymphocyte infiltration into absent, low, and moderate. However, H&E analysis cannot define specific lymphoid subpopulations unless specific immunostaining protocols are applied [28]. Therefore, to better define the immune cell components residing in the breast tissue, we decided to exploit a 10-color flow cytometry approach described hereafter.

### 3.2. Analysis of Breast-Infiltrating Immune Cells by Multicolor Flow Cytometry

The FCM analysis protocol (Figure 2 and Appendix A) was conceived to discriminate between immune cells (CD45^pos^EpCAM^neg^) and epithelial cells (CD45^neg^EpCAM^pos^) on 15 and 14 biopsies from malignant and benign breast tissues, respectively. Live cells were selected according to the physical parameters FSC and SSC, and then single cells were identified in the FSC-H vs. FSC-A dot plot (Figure 2A,B, left plot). Immune cells and epithelial cells are then identified in the CD45 vs. EpCAM dot plot (Figure 2A,B, right plot). We observed that the percentage of immune cells was significantly higher in tissues obtained from patients’ breast fibroadenomas (* *p* < 0.05) compared with breast cancer patients (Figure 2C, left plot). Conversely, no significant differences were observed regarding epithelial cells (Figure 2C, right plot). We went on to perform a deep-immunophenotyping of the CD45+ cells identified through the evaluation of the T-, B- and NK lymphocytes, as well as the monocyte subpopulations.

### 3.3. T Lymphocytes

T lymphocytes were identified as CD45^pos^CD3^pos^ (Figure 3A–D) events and were significantly increased (**** *p* < 0.0001) in tumor lesions when compared to fibroadenomas (Figure 4A and Table 2). Furthermore, we defined the percentages of helper and cytotoxic T cells according to the CD4 and CD8 expression (Figure 3E and Figure 4B,C and Table 2). We found that CD45^pos^CD3^pos^CD4^pos^ helper T cells were significantly higher in malignant breast lesions (Figure 4B and Table 2). In contrast, CD45^pos^CD3^pos^CD8^pos^ cytotoxic T cells disclosed an opposite trend: the CD4/CD8 ratio is significantly unbalanced in the direction of breast cancer patients (Figure 4D and Table 2). Moreover, the percentage of activated CD4 T lymphocytes, which was determined by evaluating those expressing the MHC class 2 membrane receptor (the Human Leukocyte Antigen DR isotype—HLA-DR,), is observed to be significantly less active in the tumor microenvironment than those present in fibroadenomas (Figure 3F,G and Figure 4E and Table 2). This finding suggests that the tumor microenvironment may affect the ability of CD4+ T lymphocytes to activate and attract cytotoxic components to the tumor site, leading to the inability of the anti-cancer immune response to function properly.

### 3.4. T-Reg

T-reg cells have been associated with the clinical outcome of breast cancer patients [29,30]. For these reasons, we analyzed the Treg populations in breast cancer tissues and in the healthy counterpart of the same subject. As shown in Figure 5, CD45^pos^/CD3^pos^/CD4^pos^/FoxP3^pos^ T-reg cells are significantly higher in tumor tissues with respect to the normal tissues (Figure 5A,B,E). Moreover, the CD45^pos^/CD3^pos^/CD4^pos^/FoxP3^pos^/Helios^pos^ T-reg subpopulation was evaluated. Figure 5C,D,F clearly show an increase in abundance of this T-reg in breast cancer tissues in comparison with the healthy breast parenchyma.

### 3.5. NK Lymphocytes

NK lymphocytes were selected according to the gating strategy shown in Figure 6. Both the NK cells (CD56^brigth^CD16^pos^) and cytotoxic NK cells (CD56^dim^CD16^pos^) were not significantly different between the two study groups (Figure 6A,B) (Figure 6A,B and Table 3). Conversely, when considering the NKT cells (CD56^pos^/CD3^pos^, Figure 6C), we found a significantly higher presence in the breast cancer patients than in the fibroadenomas (Figure 7A–C and Table 3). Similarly, type 1 NKT cells (CD56^pos^/CD3^pos^/CD4^pos^, Figure 6D) are also significantly more present in subjects with tumors than in those with fibroadenoma (Figure 7D and Table 3). The hyper-activation of NK cells and NK-T cells was also significantly higher in patients with tumors than in those with fibroadenomas (Figure 6E,F and Figure 7E,F and Table 3). In contrast, no variations in the percentage of B lymphocytes were observed between subjects with breast cancer and those with fibroadenoma (Appendix A).

### 3.6. Monocytes

Finally, the tumor-associated monocyte component was analyzed in detail compared with fibroadenoma. First of all, the lymphocyte-to-monocyte ratio was calculated by selecting all lymphocytes as CD45^pos^ cells and monocytes as CD14^pos^ cells (Figure 8A). As can be seen from Figure 9A and Table 4, the ratio is significantly higher in subjects with breast cancer, a fact also confirmed by the percentage of monocytes identified, which is significantly higher in fibroadenomas than in tumors (Figure 9B and Table 4). No significant differences were found in the percentage of atypical CD14^pos^/CD16^pos^ (Figure 8A and Figure 9C and Table 4) monocytes, or in the degree of monocyte activation (Figure 8B and Figure 9D and Table 4).

## 4. Discussion

In physiological conditions, immune cells maintain breast tissue homeostasis by continuous immunosurveillance and play a central role in initiating inflammatory reactions [31]. In the case of BC, immune cells invade the cancer tissue, and infiltrating lymphocytes play a central in cancer growth, so that the percentage of TILs is currently accepted as a useful prognostic factor for managing breast cancer patients. Histological methods (hematoxylin–eosin or immunohistochemistry staining) are used to evaluate the TILs percentage in paraffin-embedded BC tissue sections [28,32]. Although the assessment is specific to the tumor region, it does not consider the different subpopulations infiltrating the tumor, unless time-consuming and rather complex analyses are made (i.e., IHC with specific markers for each subpopulation). Identifying which subpopulations are altered in the tumor microenvironment would help design targeted intervention strategies based on immunotherapy, which has been gaining importance in the treatment of neoplasms [33]. In this scenario, our manuscript aims at studying the tumor-associated lymphocyte subpopulations in comparison to those associated with benign formations (fibroadenomas) by a multiparametric FCM approach. We performed a deep-immunophenotyping of CD45^pos^ cells through the evaluation of immune cells, with particular reference to the lymphocyte subpopulations and monocytes. According to recent observations of our research group in peripheral blood [34], we found that in the case of BC onset the tumor microenvironment featured by a higher percentage of CD4^pos^ T cells in comparison to fibroadenoma, as previously reported [35,36,37]. However, the percentage of activated CD4+ T cells was lower in case of BC malignancies. In addition, considering that CD4^pos^ T cells primarily mediate antitumor immunity by stimulating CD8^pos^ [38,39], it is not surprising that our data highlighted that the percentage of cytotoxic CD8^pos^ T cells was significantly diminished in malignant tissues. In order to assess the presence of T-reg cells in the tumor microenvironment, we performed a complex comparative analysis between the microenvironment of the tumoral lesion and that of disease-free breast tissue from the same subject. T-reg cells are rare lymphocyte subpopulations. A large number of events must be used for their characterization at the FCM. For this reason, we could determine the T-reg subpopulations only in a limited number of subjects (three tumor tissues vs. three healthy counterparts). Although preliminary, given the number of subjects analyzed, our data show that the T-reg CD45^pos^/CD3^pos^/CD4^pos^/FoxP3^pos^ and CD45^pos^/CD3^pos^/CD4^pos^/FoxP3^pos^/Helios^pos^ subpopulations are a more stable immunosuppression phenotype at the level of the tumor microenvironment and more abundant in the tumor microenvironment than in the non-diseased microenvironment, in accordance with data in the literature [30]. This evaluation shows that our approach can also be used to determine the rare subpopulations in tumor tissues.

The ability of malignant transformations to promote the onset of a cancer immunotolerance was confirmed when analyzing the NK sub-settings. These cells contribute to various immune functions during cancer initiation and progression that can change in quality and magnitude depending on the disease stage [40,41,42,43]. Our data indicate that there are no differences in the percentage of NK cells or cytotoxic NK cells (CD56^pos^/CD16^pos^) in breast cancer patients compared with fibroadenomas. On the contrary, a significant increase in NK-T cells and type I NK cells (CD56 ^pos^/CD4 ^pos^) occur in the tumor tissues. Surprisingly, NK cells and NK-T cells turn out to be more activated by the tumor microenvironment. The synergistic effect of the presence/activation of NK cells, and in particular of NK-T cells, could be in line with data in the literature, in which these effects determine the amplification or dampening of the immune response. Moreover, the hyperstimulation of NK cells during tumor progression could be associated with anergy and/or skewing of the anti-cancer immunity, which could facilitate the tumor progression and immune escape [44]. Finally, we observed a significant reduction in the percentage of monocytes in tumor tissues, resulting in an imbalance in the lymphocyte/monocyte ratio, significantly higher in subjects with tumors than in those with fibroadenomas. The tumor microenvironment reprograms monocytes by reducing their ability to respond to stimuli and assuming immunosuppressive activity [45]. Furthermore, assessing the ratio of lymphocytes to monocytes is a prognostic factor that is becoming increasingly important in the clinical management of breast cancer patients, since it is associated with a poor prognosis [46]. Furthermore, there are still unclear indications on some immune components. Conflicting current scientific data suggests that CD19+ B cells in breast cancer tissues may have positive [47], negative [48], or no significant performance [49] in humoral and cellular immunity. Furthermore, there are potential interconnections to be explored between CD19+ B cells and other lymphocytes that make the tissue microenvironment favorable or hostile to tumor progression [50].

Our approach aims to evaluate the specific subpopulations of the immune system involved in the anti-cancer response using flow cytometry. In association with normal clinical practice performed by pathological anatomy, this analysis could shed light on which cell types of the immune system are altered by the tumor microenvironment. Our approach is intended to support normal clinical practice because it could augment the information provided by routine investigations. In fact, the evaluation of TILs carried out by the pathologists provides a precise focus on the tumor portion, allowing identification of the quantity and/or quality of lymphocytes infiltrating the tumor [18,51], but does not provide a general view of the entire tissue fragment. On the contrary, our method, although it loses the ability to focus only on the region invaded by the tumor cells, provides an overall view of the entire tissue, allowing, with the same analysis, to evaluate many more parameters than a pathological anatomy investigation. In this way, truly targeted and personalized immunotherapy approaches could be developed to improve the management of breast cancer patients. For human breast tissue, isolation of single cells typically involves long mechanical and enzymatic dissociation, where the majority of the well-established protocols rely on a 16–24-h enzymatic dissociation, with varying concentrations of collagenase and/or hyaluronidase [25,27,52]. It is thus important to underline a limitation of our study. During the digestion processes, exposure to enzymatic action can lead to the disruption of cell surface markers, induction of apoptosis, and, in the worst case, a complete loss of certain immune cell subtypes [53,54]. Furthermore, comparing different enzymatic dissociation protocols, the focal technical variables, such as agitation speed, duration, and temperature, can affect the composition of the isolated cells, until inducing transcriptional changes that represents an important aspect to evaluate in gene expression studies sensitive to excessive stress caused by digestion conditions. Despite the limitations mentioned for the study of breast tumoral tissues, our approach could also be extended to other malignancies for which an understanding of the involvement of the anti-cancer immune response in the onset of the disease is still complicated. Finally, in the future, in order to better study the tumor microenvironment, other lymphocyte subpopulations (PD1, PD-L1, CTLA4, etc.) also need to be taken into account, to identify personalized approaches that are not only diagnostic but also therapeutic.

## 5. Conclusions

The study of the breast cancer microenvironment is important in the clinical management of breast cancer patients. Routine clinical practice is based on a qualitative assessment of TILs through the involvement of pathologists. Despite the innovative approaches available, such as immuno-specific IHC to study lymphocyte subpopulations, or machine learning approaches to analyze the entire histological slide, it is necessary to ensure an increasingly personalized approach to patient care. For this reason, a deep-immunophenotyping analysis performed by flow cytometry, directly on tissue recovered after biopsy, would open the way to new diagnostic and/or therapeutic approaches that could support the normal clinical routine in order to improve the management of breast cancer patients.

## Figures and Tables

**Figure 1 cancers-14-03869-f001:**
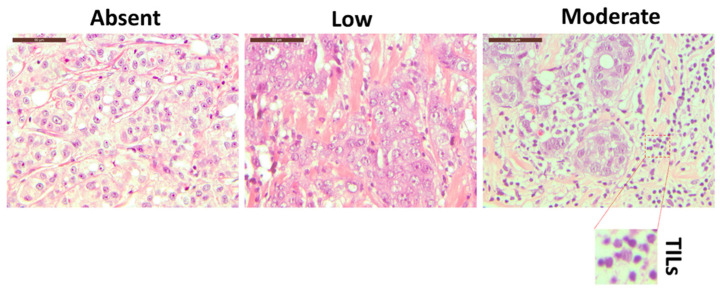
The hematoxylin–eosin staining of three different breast cancer tissue patients with three different level of TILs: absent (left panel), low (middle panel), and moderate (right panel). The inset represents an enlarged detail of the TILs. Magnification 40×. Scale bars 50 µm.

**Figure 2 cancers-14-03869-f002:**
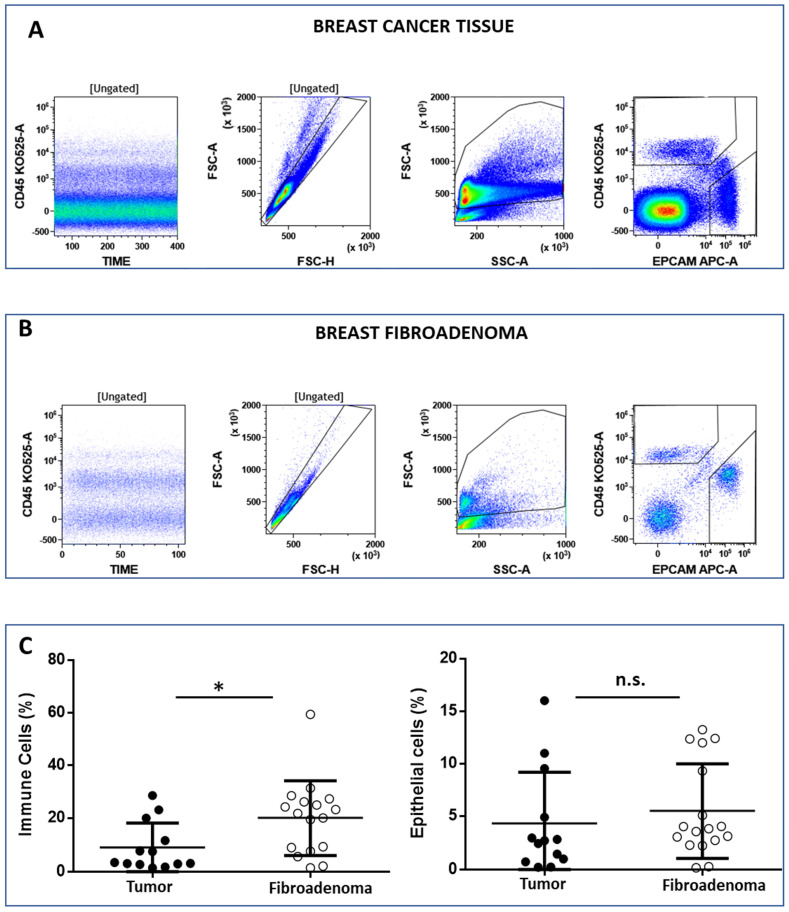
Panel (**A**) and (**B**) show the flow cytometry gating strategy to discriminate between lymphoid (CD45^pos/^EpCAM^neg^) and epithelial (CD45^neg^/EpCAM^neg^) cells. The CD45 vs. time dot plot was examined to evaluate the absence of electronic noise during the acquisition; the FSC-A vs. FSC-H plot to exclude doublets; and then the physical parameters FSC-A vs. SSC-A to select live cells and exclude debris. Panel (**C**) shows the median percentage of immune and epithelial cells in both the breast tumor and fibroadenoma samples. * = *p*-value < 0.05 unpaired *t*-test. n.s. = not significative.

**Figure 3 cancers-14-03869-f003:**
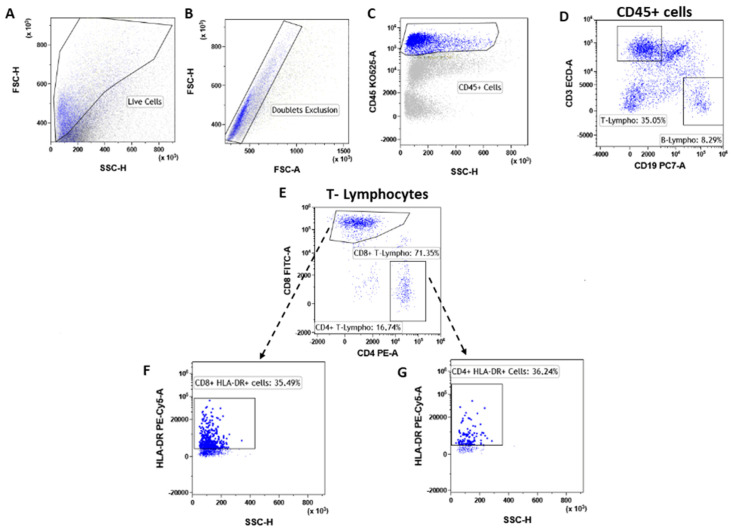
T-lymphocyte subtypes gating strategy. After doublet exclusion (**B**), the immune cells (**A**) were selected as a CD45+ event (**C**). From the CD45+ cells, we selected the T-lymphocytes as CD45^pos^/CD3^pos^ and B-lymphocytes as CD45^pos^/CD19^pos^ (**D**). Inside the T-lymphocytes we determine the CD8+ T-lymphocytes and CD4+ T-lymphocytes (**E**), which were also assessed for their level of activation by determining the expression levels of HLA-DR (**F**,**G**).

**Figure 4 cancers-14-03869-f004:**
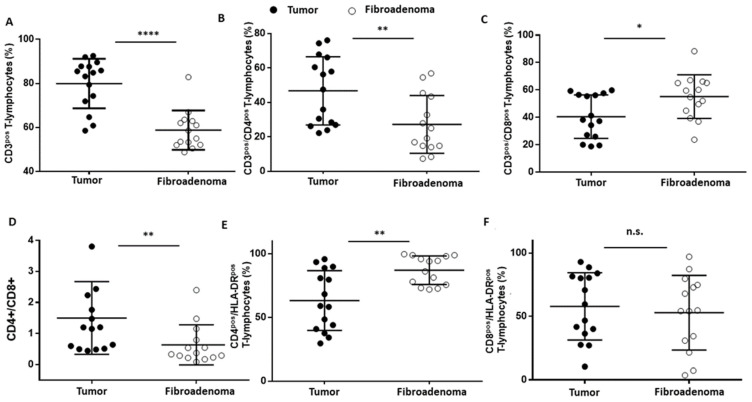
T-lymphocyte subset percentages were plotted, according to specific membrane receptor, in tumor (black dots) and fibroadenoma (white dots) tissue patients. (**A**) CD3^pos^ T-lymphocytes. (**B**) CD3^pos^/CD4^pos^ T-lymphocytes. (**C**) CD3^pos^/CD8^pos^ T-lymphocytes. (**D**) CD4^pos^-CD8^pos^ ratio. (**E**) CD4^pos^/HLA-DR^pos^ T-lymphocytes. (**F**) CD8^pos^/HLA-DR^pos^ T-lymphocytes. The median and standard deviation are reported. * = *p*-value < 0.05, ** = *p*-value < 0.01, and **** = *p*-value < 0.0001; unpaired *t*-tests. n.s. = not significant.

**Figure 5 cancers-14-03869-f005:**
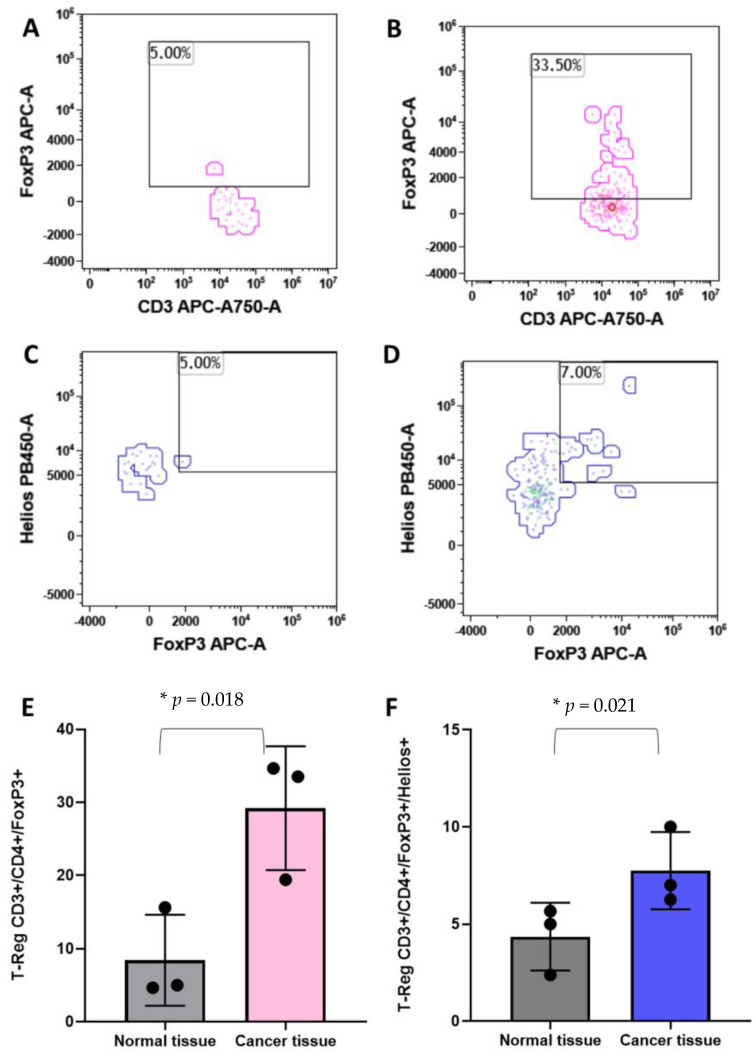
Identification of the regulatory T cells by flow cytometry. Panel (**A**) and (**B**) show the identification of T-Regs as CD45+/CD3+/CD4+/FoxP3+ events in normal and cancer tissues, respectively. Panel (**C**) and (**D**) show the T-Reg identification including the Helios markers in normal and cancer tissues, respectively. Histograms show the mean plus standard deviation of the percentage of FoxP3+ T-reg (**E**) and FoxP3+/Helios+ T-reg (**F**). T-Reg identified with the FoxP3 marker presented a mean of 8.4% (SD = 6.2%) and 29.19% (SD = 8.4%) in normal versus cancer breast tissues, respectively (**E**). The addition, the Helios marker revealed a T-Reg mean of 4.35% (SD = 1.7%) and 7.75% (SD = 1.9%) in normal versus cancer breast tissues, respectively (**F**). * = *p*-value < 0.05 unpaired *t*-test.

**Figure 6 cancers-14-03869-f006:**
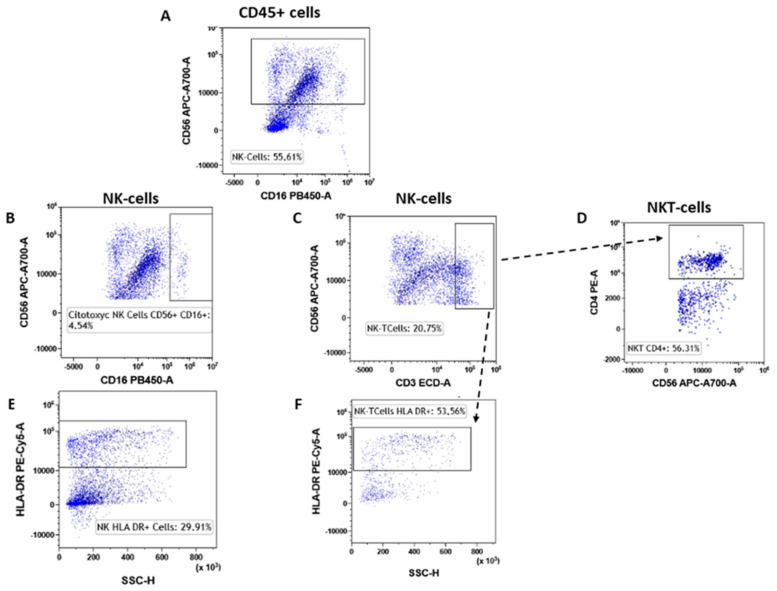
NK subtypes gating strategy. NK cells were selected starting from the CD45^pos^ event as CD45^pos^/CD16^dim^ cells (**A**). Inside the NK cells, we selected the cytotoxic NK cells (CD56^pos^/CD16^pos^, (**B**) and NK-T cells (CD56^pos^/CD3^pos^, (**C**). The NK-T type I cells were selected as CD56^pos^/CD3^pos^/CD4^pos^ (**D**). Both NK and NK-T cells were also assessed for their level of activation by determining the expression levels of HLA-DR (**E**) and (**F**).

**Figure 7 cancers-14-03869-f007:**
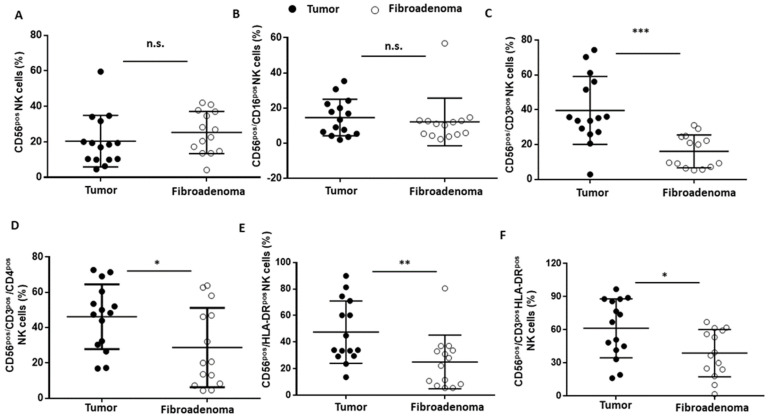
NK cell subset percentages were plotted according to the specific membrane receptor, in tumor (black dots) and fibroadenoma (white dots) tissue patients. (**A**) CD56^pos^ NK cells. (**B**) CD56^pos^/CD16^pos^ cytotoxic NK cells. (**C**) CD56^pos^/CD3^pos^ NK-T cells. (**D**) CD56^pos^/CD3^pos^/CD4^pos^ NK cells (NK Type I). (**E**) CD56^pos^/HLA-DR^pos^ NK cells. (**F**) CD56^pos^/CD3^pos^/HLA-DR^pos^ NK cells. The median and standard deviation are reported. * = *p*-value < 0.05, ** = *p*-value < 0.01, and *** = *p*-value < 0.001; unpaired *t*-tests. n.s. = not significant.

**Figure 8 cancers-14-03869-f008:**
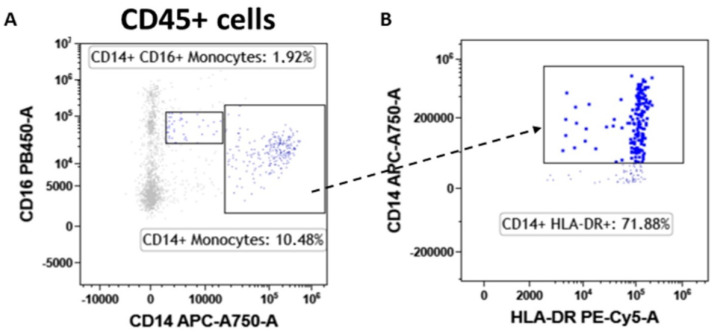
Monocyte subtype gating strategy. Inside the CD45^pos^ cells, we selected monocytes as CD14^pos^ events and atypical monocytes (CD14^pos^/CD16^pos^) (**A**). Monocytes were also assessed for their level of activation by determining the expression levels of HLA-DR (**B**).

**Figure 9 cancers-14-03869-f009:**
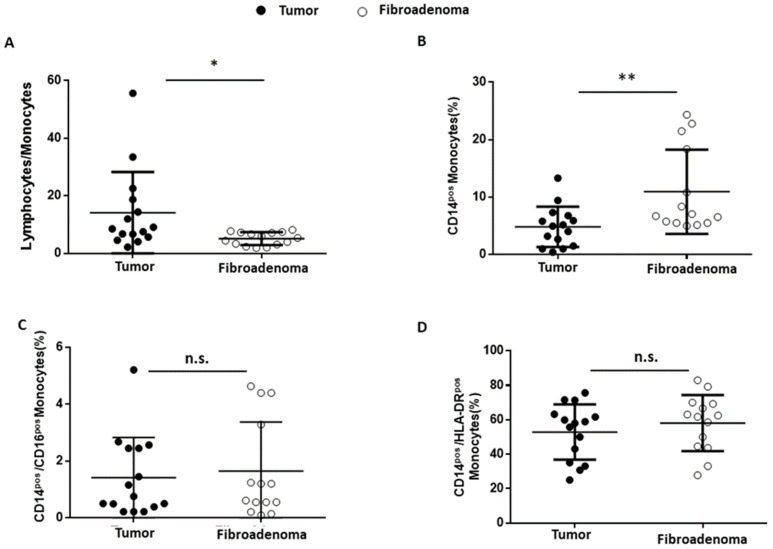
Monocyte subset percentages were plotted according to the specific membrane receptors, in tumor (black dots) and fibroadenoma (white dots) tissue patients. (**A**) Lymphocytes-to-monocytes ratio. (**B**) CD14^pos^ monocytes. (**C**) CD14^pos^/CD16^pos^ atypical monocytes. (**D**) CD14^pos^/HLA-DR^pos^ monocytes. The median and standard deviation are reported. * = *p*-value < 0.05 and ** = *p*-value < 0.01; and unpaired *t*-tests. n.s. = not significant.

**Table 1 cancers-14-03869-t001:** Clinico-pathological characteristics of the studied patients.

Fibroadenomas Sample (n = 15)		
18–50 (29)		
**Breast Cancer Sample (n=14)**		
32–77 (56)		
**Sex**		
Woman	29	
Man	0	
**Histologic Types**		
14 Invasive breast cancer “no special type”		
1 Invasive lobular carcinoma		
15 Fibroadenoma		
**Ki67**		
Low (0–29%)	11	
High (30–100%)	4	
**Grade**		
G1	3	
G2	10	
G3	2	
**Tumor size**		
	Breast malignancies	Fibroadenoma
(0.1–2 cm)	3	4
(2–5 cm)	12	11
**Breast tumor TILs grade (n)**		
Absent	Mild	Moderate
7	7	1

**Table 2 cancers-14-03869-t002:** Statistical values of the T-lymphocytes subsets. Values are reported as the median percentage of gated cells.

	T-Lymphocytes		CD4+ T-Lymphocytes		CD8+ T-Lymphocytes	
	**Tumors**	**Fibroadenoma**	**Tumors**	**Fibroadenoma**	**Tumors**	**Fibroadenoma**
**Minimum**	58.57	48.76	22.24	7.360	18.82	23.70
**25% percentile**	71.93	52.12	26.94	14.61	25.95	43.54
**Median**	84.89	56.82	47.47	22.08	38.17	57.15
**75% percentile**	87.76	63.07	66.07	43.93	56.48	65.4
**Maximum**	92.54	82.86	76.06	56.85	59.79	88.27
	**CD4+-CD8+ Ratio**		**CD4+/HLA-DR+** **T-Lymphocytes**		**CD8+/HLA-DR+** **T-Lymphocytes**	
	**Tumors**	**Fibroadenoma**	**Tumors**	**Fibroadenoma**	**Tumors**	**Fibroadenoma**
**Minimum**	0.4418	0.08338	29.52	71.92	10.4	3.61
**25% percentile**	0.511	0.2249	40.83	74.94	36.33	28.52
**Median**	1.195	0.3511	58.98	90.06	59.53	54.41
**75% percentile**	2.23	0.8828	88.79	98.27	81.54	75.97
**Maximum**	4.041	2.399	95.73	99.73	92.91	96.99

**Table 3 cancers-14-03869-t003:** Statistical values of the NK lymphocyte subsets. Values are reported as the median percentage of gated cells.

	NK cells		CD56+/CD16+ NK Cells		CD56+/CD3+ NK-T Cells	
	**Tumors**	**Fibroadenoma**	**Tumors**	**Fibroadenoma**	**Tumors**	**Fibroadenoma**
**Minimum**	4.56	4.18	2.00	2.38	2.84	5.28
**25% percentile**	9.95	14.48	5.45	4.838	27.24	6.98
**Median**	18.47	24.76	13.43	10.69	35.26	14.8
**75% percentile**	31.7	37.19	22.39	12.88	56.15	24.56
**Maximum**	59.6	42.01	35.38	56.84	74.39	31.08
	**CD56+/CD3+/CD4+** **NK cells**		**CD56+/HLA-DR+** **NK cells**		**CD56+/CD3+/HLA-DR+** **NK cells**	
	**Tumors**	**Fibroadenoma**	**Tumors**	**Fibroadenoma**	**Tumors**	**Fibroadenoma**
**Minimum**	16.88	4.35	13.71	5.56	16.22	1.790
**25% percentile**	30.3	8.053	29.53	8.15	41.52	22.61
**Median**	48.29	20.35	33.94	25.19	66.94	38.04
**75% percentile**	60.48	49.69	71.04	34.56	87.88	59.92
**Maximum**	72.73	63.86	89.91	80.45	96.88	67.00

**Table 4 cancers-14-03869-t004:** Statistical values of the monocyte subsets. Values are reported as the median percentage of gated cells.

	Lymphocytes/Monocytes Ratio		CD14+ Monocytes	
	**Tumors**	**Fibroadenoma**	**Tumors**	**Fibroadenoma**
**Minimum**	2.175	1.863	0.4300	5.000
**25% percentile**	5.656	3.074	1.510	5.515
**Median**	8.52	5.377	4.970	6.890
**75% percentile**	18.72	7.265	6.790	19.14
**Maximum**	55.64	8.158	13.32	24.34
	**CD14+/CD16+ Monocytes**		**CD14+/HLA-DR+** **Monocytes**	
	**Tumors**	**Fibroadenoma**	**Tumors**	**Fibroadenoma**
**Minimum**	0.2200	0.0900	25.00	27.83
**25% percentile**	0.3900	0.4650	35.11	44.38
**Median**	0.7600	0.9050	58.02	62.14
**75% percentile**	2.450	3.568	63.27	69.47
**Maximum**	5.220	4.640	75.68	83.02

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
