# Peer review of "Identification of Immune Cell Components in Breast Tissues by a Multiparametric Flow Cytometry Approach"

_cancers, 2022, doi:10.3390/cancers14163869_

Round 1

Reviewer 1 Report

To authors:

The manuscript by Coppola et al, “Identification of immune cell components in breast tissues by a multiparametric flow cytometry approach” is a study about comparing breast cancer tissue with fibroadenoma tissue using a multiparametric flow cytometry approach with focus on tumor infiltrating lymphocytes (TILs).

The authors claim that the analysis today of the tumor microenvironment and TILs in breast cancer are evaluated in classical H&E stained histological specimens by experienced pathologists, and that other techniques such as specific IHC staining can be more affordable to define TILs immune components. However, the role of IHC and references stating the importance and development is not included. Instead, the authors present their approach using multiparametric flow cytometry approach with focus on tumor infiltrating lymphocytes (TILs).

I have several major comments:

Introduction

-The introduction starts out with describing gene expression. This is a bit misleading since the paper is not about gene expression at all, but about protein TIL biomarkers. It would be interesting to read more about the role of the infiltrating immune cells in breast cancer.

-The sentence on line 47-50 states the importance of the breast cancer microenvironment as a prognostic factor to evaluate possible potential therapeutic targets, but it does not tell how this can be used.

-References 8 and 9 that are used to claim that “in the case of breast cancer, a more favorable clinical outcome was reported when higher numbers of TILs were detected in hematoxylin and eosin (H&E) stained histological slides via light microscopy” are from The International Immuno-Oncology Biomarker Working Group and a conference proceedings, respectively. The authors should in addition use original peer-reviewed publications to state this.

-The actual immune cells analysed in this paper is confusing and not clearly described since in the title “immune cell components” is stated, in the abstract “TILs” is stated, but in the introduction and lower part of the abstract “TILs, NK cells, and monocytes” and “Cytotoxic (CD8+) T cells, together with varying proportions of helper (CD4+) T cells and CD19+ B cells, and rare NK cells are the principal components of TILs.”, respectively, is stated. Moreover, the analysis of CD14+ monocytes is a part of the result section. The authors should be more clear about the actual cell types analysed and describe this in the abstract and introduction.

-Lines 83-90 are lacking references. Please include actual novel references.

-There is no description in the introduction of the approach for cell analysis from solid tissue.

Material and methods

-Are there any references that can be included in the methods section? Have the method preparing and using tissue for flow cytometry been used and published by the authors? Have the use of the antibodies described for this method been published by the authors or by others?

Result

-The authors claim that “H&E analysis cannot define specific lymphoid subpopulations unless specific immunostaining protocols are applied. Therefore, to better define the immune cell components residing in the breast tissue, we decided to exploit a 10-color flow cytometry approach described hereafter”. However, the possible IHC approach is not mentioned at all. Did the authors perform any IHC? If not, can you justify that?

-The inset in Figure 1, representing an enlarged detail of TILs is not sharp. Can the authors also describe how the TILs were identified? What is the difference between what is presented in Figure 1 and Supplementary Figure 1??

Discussion

References are needed to support the authors statements:

-Lines 262-267 are lacking references. Please include actual novel references.

-Lines 274-279 are lacking additional references. Please include actual novel references.

-Lines 303-310 are lacking references. Please include actual novel references.

In addition:

-The discussion about CD19+ B cells is missing.

-A deep discussion about the method and approach with appropriate references is missing.

Reviewer 2 Report

In this manuscript, Coppola et al. investigated the immune cell landscape in samples from breast cancer and fibroadenoma patients. The authors used mechanical disruption and enzymatical digestion to obtain single cell suspensions of tumor samples, followed by antibody staining and flow cytometric analyses. Although the reported differences between breast cancer and fibroadenoma samples are interesting, the originality and potential implications of this study is limited. The authors describe variations in the main immune cell populations, but a comprehensive and deep analysis of immune phenotypes is missing. The study does not address fundamental questions, such as what are the specific molecular and functional characteristics of the immune cell subsets. For instance, increased CD4 T cell frequencies in breast cancer samples is interesting, but without further classification in conventional or regulatory T cells no conclusion can be drawn. In order to get an impression of the immune landscape and potential cell interactions, the authors should characterize in detail the suppressive cells (Treg, MDSC,…) and effector cells (CD4 subsets, CD8, NK) and effector molecules (Granzyme, Perforin,…); in addition expression of inhibitory molecules (PD1, CTLA4, LAG3,…) would further strengthen the study and provides possible connection points for using checkpoint antibody therapy.

In addition to these thematic difficulties, there are other issues to be improved. The graphs and legends often lack the description of the cell population to which the frequencies refer (% of leukocytes? % of events? % of T cells?). Pie chart in Fig. 9 is unclear: what is t-lympho? Are CD4+ and activated CD4+ two different populations? In Fig 8 lymphocyte-to monocytes ratio is shown. How are lymphocytes determined? The gating strategy in Fig. 3 (FCS-H/SSC-H,…) differs from Fig. 2. PI staining to discriminate live/dead cells as mentioned in M&M is missing. The manuscript has to be corrected in grammar and syntax. In Fig. 5, a big fraction of CD56int/CD16int cells is shown and the CD56 gate is arbitrarily placed.  

Round 2

Reviewer 1 Report

The manuscript by Coppola et al, “Identification of immune cell components in breast tissues by a multiparametric flow cytometry approach” have been revised according to the suggestions. I am satisfied with this and have no more comments.

Author Response

We would like to thank the reviewer very much for the precise and helpful comments that enabled us to improve the manuscript

Reviewer 2 Report

In this manuscript, Coppola et al. investigated the immune cell landscape in samples from breast cancer and fibroadenoma patients. Most of my questions were answered and analysis of Treg cells were added. However, these analyses were restricted to tumor samples and healthy tissue from 3 patients. All other figures compare breast cancer vs. fibroadenoma. In addition, gating strategy and FMO controls for Treg cell analyses are missing. Finally, the graphs and legends still lack the exact description of the cell population to which the frequencies refer. This should be clearly indicated in the y-axis (e.g. % of NK cells).
